# Conformational changes in the catalytic region are responsible for heat-induced activation of hyperthermophilic homoserine dehydrogenase

Tatsuya Kubota[1], Erika Kurihara[2], Kazuya Watanabe[1], Kohei Ogata[2], Ryosuke Kaneko[2], Masaru Goto[2], Toshihisa Ohshima[3] & Kazuaki Yoshimune[1✉]

When overexpressed as an immature enzyme in the mesophilic bacterium *Escherichia coli*, recombinant homoserine dehydrogenase from the hyperthermophilic archaeon *Sulfurisphaera tokodaii* (StHSD) was markedly activated by heat treatment. Both the apo- and holo-forms of the immature enzyme were successively crystallized, and the two structures were determined. Comparison among the structures of the immature enzyme and previously reported structures of mature enzymes revealed that a conformational change in a flexible part (residues 160–190) of the enzyme, which encloses substrates within the substrate-binding pocket, is smaller in the immature enzyme. The immature enzyme, but not the mature enzyme, formed a complex that included $NADP^+$, despite its absence during crystallization. This indicates that the opening to the substrate-binding pocket in the immature enzyme is not sufficient for substrate-binding, efficient catalytic turnover or release of $NADP^+$. Thus, specific conformational changes within the catalytic region appear to be responsible for heat-induced activation.

---

[1] Department of Applied Molecular Chemistry, Graduate School of Industrial Technology, Nihon University, 1-2-1, Izumichou, Narashino, Chiba 275-8575, Japan. [2] Department of Biomolecular Science, Graduate School of Science, Toho University, 2-2-1, Miyama, Funabashi, Chiba 274-8510, Japan. [3] Department of Biomedical Engineering, Osaka Institute of Technology, 5-16-1, Ohmiya, Asahi-ku, Osaka 535-8585, Japan. ✉email: yoshimune.kazuaki@nihon-u.ac.jp

Enzymes from hyperthermophiles are adapted to high temperatures[1]. Some highly thermophilic enzymes over-expressed as recombinant proteins in mesophiles, such as *Escherichia coli*, are produced in an immature form that exhibits little or no activity. For example, recombinant glutamate dehydrogenases (GDHs) from the hyperthermophiles *Pyrobaculum calidifontis*[2] and *Pyrococcus furiosus*[3] are known to be expressed as immature forms in *E. coli*. However, they can be transformed to their mature form by incubation at a high temperature that is near the growth temperature of hyperthermophiles. The high temperature is thought to promote proper folding or oligomeric structure formation by these enzymes from hyperthermophiles. The mechanism underlying heat-induced maturation has been investigated using differential scanning microcalorimetry[4], X-ray small-angle scattering diffraction[5,6] and circular dichroism[4], as well as by studying the enzymes' catalytic activity. However, since the crystal structures of the immature enzymes have not yet been determined, details of the three-dimensional structural basis of the mechanism remain unclear.

Homoserine dehydrogenase (HSD) catalyzes NAD(P)H-dependent production of homoserine from L-aspartate-4-semialdehyde and functions in the aspartate pathway, catalyzing the synthesis of cysteine, threonine and methionine from aspartate in plants, fungi, yeast, bacteria and archaea[7]. We previously showed that heat-treated HSD from the hyperthermophilic archaeon *Sulfurisphaera tokodaii* strain 7 (StHSD) consists of nucleotide-binding (residues 1–130 and 285–304), dimerization (residues 131–145 and 256–284), and catalytic (residues 146–255) regions and that it is activated via reductive cleavage of the disulfide bond between cysteine residues (C303) in the C-terminal nucleotide-binding regions of the subunits forming the homodimer[8]. Moreover, our analysis of the crystal structures of the apo-form of mature StHSD (referred to as the M structure; Protein Data Bank (PDB) entry 4YDR)[8] and its holo-form complexed with NAD+ and cysteine (referred to as the M/NAD/Cys structure; PDB entry 5X9D)[9] revealed that StHSD is inhibited by cysteine through formation of an enzyme-NAD+-cysteine complex with a $K_i$ of 11 μM[9]. HSD from another hyperthermophile, *Pyrococcus horikoshii* (PhHSD), strongly binds NADP+ with a $K_i$ of 5.2 nM[10]. The crystal structure of PhHSD showed it to be complexed with NADPH despite the absence of cofactors, such as NADP+ or NADPH, in the crystallization solution. In addition to the crystal structures of PhHSD and StHSD, those of HSDs from four other thermophilic strains, *Thermoplasma volcanium* (PDB 3C8M and 3JSA), *Thermoplasma acidophilum* (PDB 3ING), *Archaeoglobus fulgidus* (PDB 3DO5) and *Thermus thermophilus* (PDB 2EJW and 6A0R), have also been deposited in the PDB. The active site residues are well conserved among the amino acid sequences of these HSDs.

In the present study, we investigated the maturation of recombinant HSD from *S. tokodaii*, which optimally grows at 353 K[11] but was overproduced in *E. coli* at 310 K. Using the structural information from the immature enzyme in its substrate-free and substrate-bound forms, the structural features responsible for heat-induced activation of the enzyme are discussed.

## Results and discussion

**Activation by heat treatment**. Recombinant StHSD was overproduced in *E. coli* grown at 310 K. The activity of the crude extract (2.0 U/mL, 0.18 U/mg) was increased approximately 2.1 times (4.2 U/mL, 4.7 U/mg) by heat-treatment (343 K for 2 h), which precipitated the major proteins from *E. coli*. The immature StHSD was purified using TOYOPEARL DEAE-650M and Butyl-650M (Tosoh Corp.) columns without heat treatment. From a structural analysis of the purified enzyme, binding of NADP+

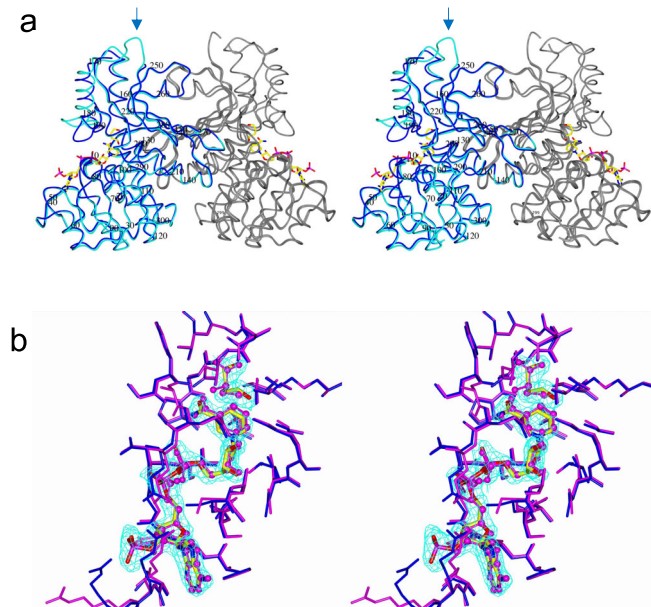

**Fig. 1 The IM and IM/NADP/BU structures in their physiological dimeric form. a** The IM and IM/NADP/BU structures are shown in cyan and blue, respectively, with the other subunit in gray. NADP+ atoms are shown as cylinders. The arrow indicates a disordered part (227–231 aa). **b** The substrate-binding pockets of the IM/NADP/BU and M/NAD/Cys structures are shown in blue and pink, respectively. NADP+ and BU are shown as cylinders, while NAD+ and Cys are shown as ball and stick. The 2Fo–Fc electron density in the map is contoured at the +1.0 sigma level.

and 1,4-butanediol (BU) to the enzyme was detected, even though NADP+ was not added during the crystallization (Fig. 1). BU was found at the binding site for homoserine (Fig. 1b). This structure was designated the IM/NADP/BU structure. An earlier report showed that NAD(P)+ is absent from the M structure formed by heat-treatment at 343 K for 3 h before purification[8].

The immature enzyme was further purified using a Blue Sepharose column which is an affinity column for purification of enzymes requiring adenyl group-containing molecules, such as NAD(P)+ and ATP. Since StHSD binds to a Blue Sepharose column instead of NAD(P)+, the enzyme in the eluant after binding to the column should be the apo-form. The absence of NAD(P)+ from the purified enzyme was also confirmed by structural analysis of the immature enzyme. The structure of the immature apoenzyme was designated the IM structure. The mature enzyme showed significantly higher specific activity at alkaline pHs (Fig. 2) at 303 K where both enzymes are stable during the assay (5 min) and showed higher specific activity at all of the tested temperatures (Fig. 3) as compared to those of immature enzyme. $V_{max}$ of the mature enzyme was approximately two times higher than that of the immature enzyme at pH 8.0 and 303 K (Table 1).

**Conformational changes**. The mature (M and M/NAD/Cys) and immature (IM and IM/NADP/BU) structures were compared to determine the effect of heat-induced maturation on the structures (Fig. 4). The M/NAD/Cys structure has structurally unique residues 22–26, mainly due to their contact with other molecules within the crystal which belongs to space group $I4_1$, and peak signals are seen by the comparison (Fig. 4b, 5b) with the other three structures solved in space group $P2_1$.

Heat treatment-induced conformational changes were more obvious in a region of residues 181–190 in the apo-form (Fig. 4a) than those observed in the substrate analog-bound forms (Fig. 4b).

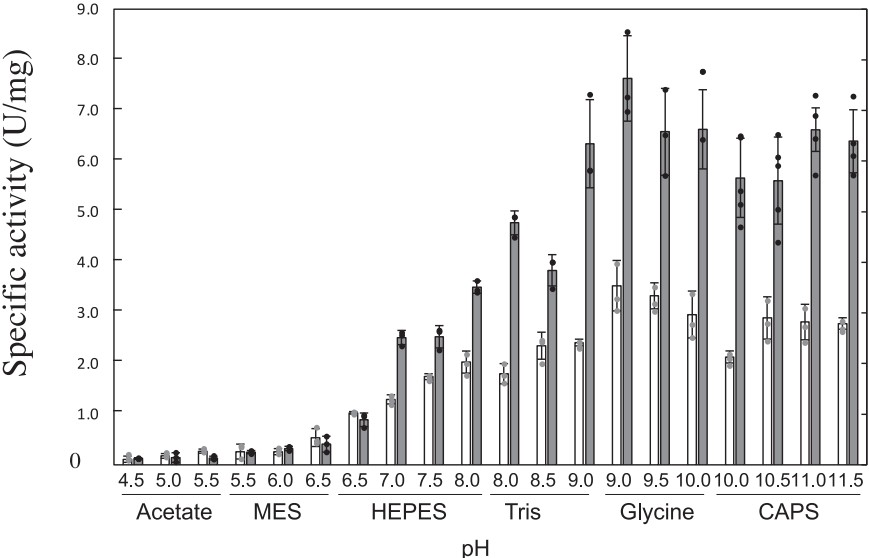

**Fig. 2 Effect of pH on enzyme activity.** The activity of the immature (white bar) and mature enzymes (gray bar) were assayed in 100 mM acetate (pHs 4.5, 5,0 and 5.5), MES (pHs 5.5, 6.0 and 6.5), HEPES (pHs 6.5, 7.0, 7.5 and 8.0), Tris (pHs 8.0, 8.5 and 9.0), Glycine (pHs 9.0, 9.5 and 10.0) or CAPS (pHs 10.0, 10.5, 11.0 and 11.5) at 303 K. The length of the error bars represents the standard deviation.

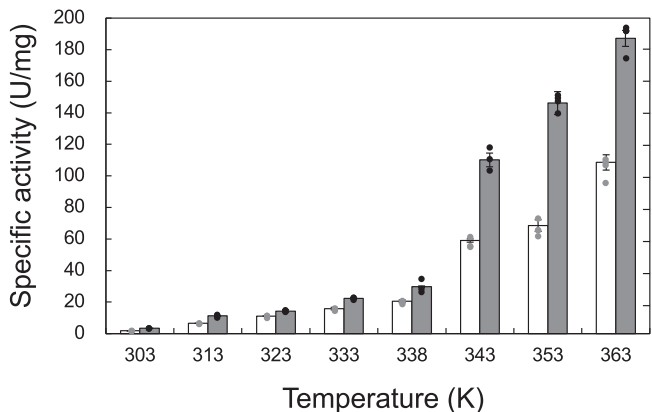

**Fig. 3 Effect of temperature on enzyme activity.** The activity of the immature (white bar) and mature enzymes (gray bar) were assayed in 100 mM tris (pH 8.0) at various temperatures. The length of the error bars represents the standard deviation.

Although the substrate analogs were distinct in the M/NAD/Cys and IM/NADP/BU structures, they share similar structures (Fig. 4b), except for the residues having peaks at 23 due to crystal packing. This suggests that these substrate analogs induce similar conformational change of StHSD by their binding.

The catalytic region (residues 146–255) changed its conformation upon NAD(P)$^+$ binding (Fig. 5). This conformational change induced by ligand binding was smaller in the immature enzyme (Fig. 5a) than that in the mature enzyme (Fig. 5b). A part (residues 160–190) of the catalytic region showed greater flexibility in its responses to heat treatment and NAD(P)$^+$ binding, and was therefore designated the flexible part. Because conformational changes induced by NAD(P)$^+$ binding to the flexible part were smaller in the immature enzyme than in the mature enzymes (Fig. 5), heat treatment was able to induce a larger conformational change in that region. The flexible part (residues 160–190) is located at the entrance of the substrate-binding pocket (Fig. 6a). In the immature apoenzyme, the substrate-binding pocket assumes a more closed form than in the mature apoenzyme. The more opened form of the substrate-

binding pocket in the mature enzyme can accelerate the catalytic turnover that results from its activation. In addition, this impediment for the catalytic turnover of the immature enzyme could stabilize ligand-binding form to show the significantly lower $K_m$ values as compared to those of the mature enzyme (Table 1).

The immature structures tend to have more disordered residues than their mature counter parts (Supplementary Table 1). Increase of folding rate of protein L at higher temperature[12] supports a hypothesis that heat-treatment accelerates proper folding of StHSD. Disordered parts in the IM/NADP/BU structure are found in the flexible part or a part adjacent to the flexible part (Fig. 1A). It is possible that their stabilization leads to the heat-induced activation by facilitating conformational change of the flexible part. On the other hands, the immature enzyme which has more closed form than in the mature apoenzyme may have higher affinity to NAD(P)$^+$ (Table 2 and Supplementary Table 2).

**The flexible part.** As mentioned above, the flexible part is situated within the catalytic region. Alignment of the stable α-helix structure (residues 194–220) adjacent to the flexible part revealed the displacement of the flexible part (Fig. 6b). The flexible part includes two α-helices and a β-turn harboring Glu186, which encloses the bound NAD(P)$^+$. Alignment of the area that includes the two α-helices and β-turn (residues 160–180) showed that this area is rigid and was therefore designated the rigid part (Fig. 6c). The area adjacent to the region that includes the β-turn structure in the rigid part was designated the lid part (residues 181–190). Upon NAD(P)$^+$ binding, the lid part interacts with the rigid part and changes its location substantially to form a hydrogen bond between E186 and R38. The distance between the rigid and lid parts within the IM structure (cyan in Fig. 6c) is larger than within the M structure (represented in pink in Fig. 6c). Both hydrophobic interactions and hydrogen bonds are thought to strengthen the interaction between the lid and rigid parts. The hydrophobic effect increases with increasing temperature, and hydrophobic interactions are known to play major roles in strengthening of thermostability of thermophilic enzymes[13–15]. This strengthened interaction may explain why the substrate-binding pocket is more open in the M structure than

**Table 1 Kinetic parameters of immature and mature enzymes.**

|  | NAD$^+$ | | Homoserine | |
|---|---|---|---|---|
|  | $V_{max}$ (U/mg) | $K_m$ (mM) | $V_{max}$ (U/mg) | $K_m$ (mM) |
| Immature | 2.3 ± 0.062 | 0.25 ± 0.027 | 2.4 ± 0.2 | 0.082 ± 0.03 |
| Mature | 5.5 ± 0.14 | 1.1 ± 0.13 | 5.0 ± 0.38 | 0.49 ± 0.05 |

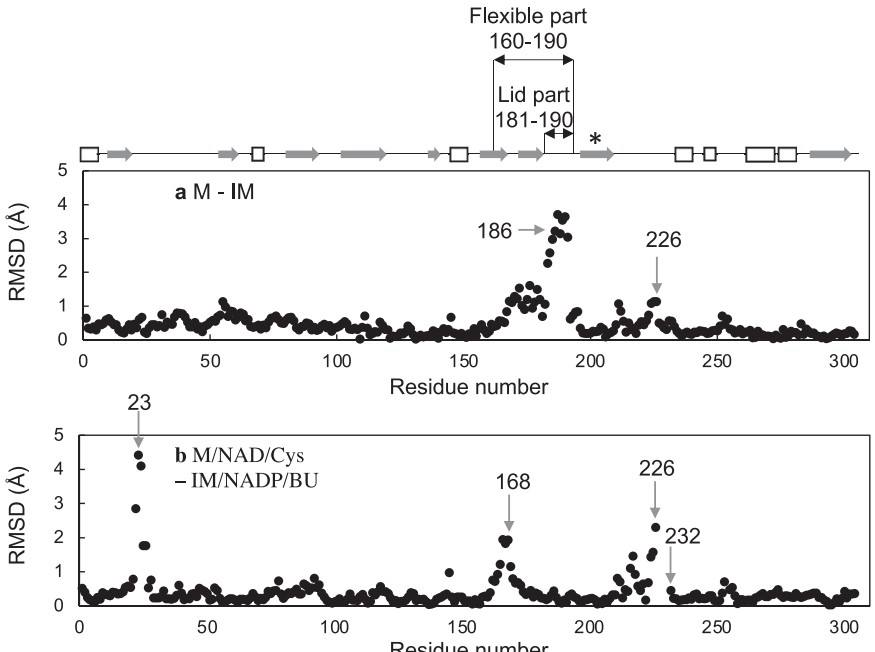

**Fig. 4 Conformational change induced by heat-treatment.** Differences between the backbone atoms of the M and IM (**a**) and M/NAD/Cys and IM/NADP/BU (**b**) structures are superimposed. The arrows and numbers indicate the residues mentioned in the text. The flexible (160–190 aa) and lid part (181–190 aa) are highlighted by two-way arrow. The helix structure (194–210 aa) described in Fig. 6 is marked with an asterisk.

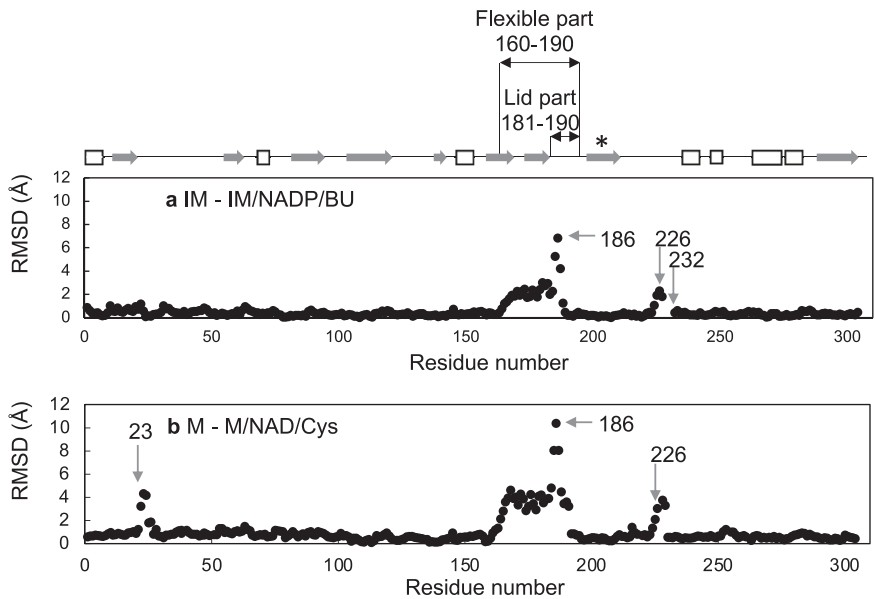

**Fig. 5 Conformational change induced by ligand binding.** Differences between the backbone atoms of the IM and IM/NADP/BU (**a**) and M and M/NAD/Cys (**b**) structures are superimposed. The arrows and numbers indicate the residues mentioned in the text. The a-helix and b-sheet structures are shown in arrows and white boxes, respectively. The flexible (160–190 aa) and lid part (181–190 aa) are shown by two-way arrow. The helix structure (194–210 aa) described in Fig. 6 is marked with an asterisk.

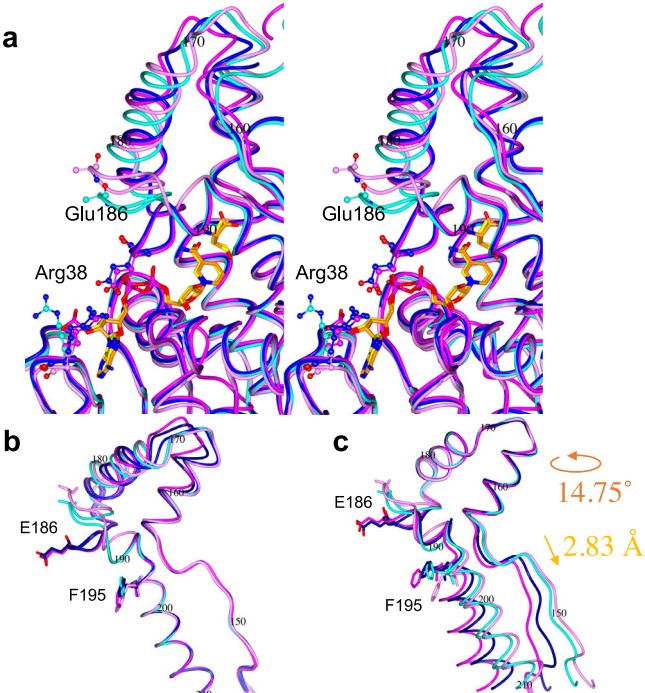

**Fig. 6 Comparison of substrate-binding pockets. a** The substrate-binding pockets within the IM (cyan), IM/NADP/BU (blue), M (pink) and M/NAD/Cys (magenta) structures are shown. The atoms of Glu186 and Arg38 are shown as ball and stick. The substrates and substrate analogs of the IM/NADP/BU (yellow) and M/NAD/Cys (orange) structures are shown as cylinders. Side-chain atoms of Glu186 within the IM and M structures are disordered. The backbone atoms of the helix structure (194–210 aa) (**b**) or the rigid part (residues 160–180) (**c**) of the IM (cyan), IM/NADP/BU (blue), M (pink) and M/NAD/Cys (magenta) structures are superposed. Superposition of the rigid parts of the IM and M structures (**b**) requires 14.75° rotation and 2.83 Å displacement. The side chains of residues E186 and F195 are shown as cylinders.

the IM structure (Fig. 6a). In addition, the side chain of F195, which is adjacent to the lid part, may interfere with effective opening of the substrate-binding pocket in the immature enzyme. The location of F195 within the immature enzyme was slightly altered by substrate binding, whereas it was markedly altered in the mature enzyme (Fig. 6c).

**NADP$^+$ binding**. Arg38 forms two hydrogen bonds with Glu186 to enclose the bound NAD(P)$^+$ within the ligand-bound structures (the IM/NADP/BU and M/NAD/Cys structures) (Fig. 7). These hydrogen bonds appear to stabilize the closed state of StHSD (the IM/NADP/BU structure) and support tight binding of NAD(P)$^+$ within the nucleotide-binding pocket. The side chains of Arg38 and Arg39 form hydrogen bonds with the phosphate group at the 2′- position of the ribose moiety of NADP$^+$ (Fig. 7), but not NAD$^+$, within the ligand-bound structures. This tight binding of NADP$^+$ can restrain turnover, resulting in the inactivation of the NADP$^+$-bound immature enzyme. Both the immature and mature enzymes were competitively inhibited by NADP$^+$ with apparent $K_i$ of 26 ± 1.7 and 48 ± 1.2 μM, respectively at pH 7.0 at 303 K (Supplementary Table 2). This low apparent $K_i$ can explain that the immature enzyme forming a complex with NADP$^+$ probably derived from the host *E. coli* cells.

HSD from *P. horikoshii* is also strongly inhibited by NADP$^+$, and the crystal structure of the enzyme shows the presence of

**Table 2 Data collection and refinement statistics (molecular replacement).**

|  | IM/NADP/BU | IM |
|---|---|---|
| **Data collection** |  |  |
| Space group | $P2_1$ | $P2_1$ |
| Cell dimensions |  |  |
| *a, b, c* (Å) | 55.33, 79.05, 65.91 | 57.1, 78.31, 65.55 |
| α, β, γ (°) | 90, 106.51, 90 | 90, 106.17, 90 |
| Resolution (Å) | 44.05–1.90 | 48.59–2.05 |
|  | (2.00–1.90) | (2.16–2.05) |
| $R_{merge}$ | 6.7 (28.1) | 6.8 (57.9) |
| $I/\sigma I$ | 11.8 (3.4) | 10.9 (2.2) |
| Completeness (%) | 98.4 (91.3) | 99.7 (99.9) |
| Redundancy | 3.9 (3.3) | 3.7 (3.8) |
| **Refinement** |  |  |
| Resolution (Å) | 49.36–1.90 | 49.59–2.05 |
| Reflections unique | 42,219 | 34,762 |
| $R_{work}/R_{free}$ | 0.211/0.257 | 0.231/0.281 |
| No. atoms |  |  |
| Protein | 4295 | 4457 |
| Ligand/ion | 108 | 2 |
| Water | 238 | 164 |
| $B$-factors(Å$^2$) |  |  |
| Protein (Chain A/B) | 27.73/31.84 | 44.478/48.302 |
| Ligand/ion (Chain A/B) | 28.18/38.18 | 45.93 (Mg$^{2+}$)/− |
| Water | 38.12 | 47.07 |
| R.m.s. deviations |  |  |
| Bond lengths (Å) | 0.0085 | 0.0098 |
| Bond angles (°) | 1.3878 | 1.4362 |

Single crystal was used for each data collection. Values in parentheses are for highest-resolution shell.

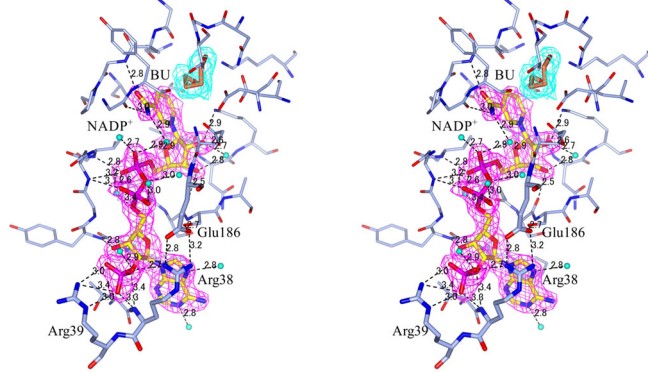

**Fig. 7 Stereo-view of a part of the catalytic region.** Hydrogen bonds within the substrate-binding pocket of the IM/NADP/BU structure are shown. The 2Fo–Fc electron density in the map is contoured at the +1.0 sigma level.

NADP$^+$ despite its absence during crystallization as does StHSD[16]. In HSD from *Thermus thermophilus*, Arg44 and Ala181 respectively correspond to Arg38 and Glu186 in StHSD, but no corresponding hydrogen bond is found[16]. Nor was it found in HSDs from *Thermoplasma volcanium* (PDB 3JSA: NAD$^+$ complex) or *Thermoplasma acidophilum* (PDB 3ING: NAD$^+$ complex), judging from the ligand-bound structures.

**Disulfide bond of the structures**. The mature enzyme is activated via reductive cleavage of the disulfide bond between the respective cysteine (C303) residues in the C-terminal nucleotide-binding region of each subunit of the homodimeric enzyme[8]. The immature enzyme was less stable than the mature enzyme in the presence of 10 mM DTT or 2-mercaptoethanol.

The immature enzyme decreased its activity approximately half of the initial activity after 12 h incubation at 277 K, though the mature enzyme increased the activity approximately 1.2 times at the same conditions. The immature enzyme probably requires the disulfide bond for the stability due to the inappropriate folding especially in the flexible part. All the structures obtained here had a disulfide bond between the C303 residues in the C-terminal regions of the two subunits. The distance between the two sulfur atoms in both the IM and IM/NADP/BU structures was 2.1 Å, which is similar to that in the oxidized enzyme (the M structure; 2.2 Å) but less than that in the reduced enzyme (2.9 Å)[8].

## Methods

**Purification**. *E. coli* BL21 strain was transformed with pSTHSD and recombinant StHSD was overexpressed at 310 K after 3 h induction by 0.1 mM isopropyl β-D-1-thiogalactopyranoside before harvesting culture[8]. The crude extract prepared from 12 g (wet weight) of *E. coli* cells was centrifuged for 20 min at 10,000 *g* and 277 K. The supernatant was dialyzed against 10 mM Tris-HCl (pH 8.5) and was then applied to a DEAE-TOYOPEARL column. The column was washed with 10 mM Tris-HCl (pH 8.5) containing 50 mM NaCl, and the fractions were eluted with 10 mM Tris-HCl (pH 8.5) containing 75 mM NaCl. The active fractions were collected and applied to a Butyl-TOYOPEARL column in the presence of 40% saturated ammonium sulfate in 10 mM Tris-HCl (pH 8.0). The enzyme was eluted with a gradient of 40 to 0% saturated ammonium sulfate. For purification of the apo-form without NADP⁺, the purified enzyme was applied to a Blue Sepharose column, and the apoenzyme was eluted with a gradient of 50 mM to 1 M NaCl in 10 mM Tris-HCl (pH 8.5). The purified enzyme (about 14 mg) was dialyzed against 10 mM phosphate buffer (pH 7.0) and concentrated using an Amicon Ultra 10 K filter unit (Millipore). The homogeneity of the final preparation was confirmed with SDS-PAGE.

**Enzyme and protein assay**. HSD activity was determined in a reaction mixture containing 0.1 M Tris-HCl buffer (pH 8.0), 10 mM L-homoserine, and 10 mM NAD⁺ and the enzyme. Activity was assessed based on the initial velocity of NADH production measured at 340 nm (molar extinction coefficient of 6220 cm$^{-1}$M$^{-1}$) during the first 120 s of incubation. The standard assay conditions were carried out at low temperature (303 K) at which the immature and mature enzymes are stable at pHs from 4.5 to 11.0 for 5 min. The immature and mature enzymes were partially inactivated after heat treatment at 353 K for 10 min at some pHs (Supplementary Fig. 1). Protein concentrations were measured using a Pierce® BCA protein assay kit (Thermo Scientific Pierce). One unit of enzyme activity was defined as the amount of enzyme that catalyzed the formation of 1 μmol NADH per minute. The types of inhibition were determined using Lineweaver-Burk plots. The $V_{max}$ and $K_m$, values and the apparent $K_i$ values were calculated by fitting Michaelis-Menten (Supplementary Fig. 2) and Morrison (Supplementary Fig. 3) equations, respectively, to the assay data using Solver in Excel.

**Structural determination**. Crystals of the IM/NADP/BU and IM structures were grown under similar conditions used for the M structures at 295 K. Crystals for the IM/NADP/BU structure were obtained in a protein solution consisting of purified enzyme (9.5 mg/mL), 20% [w/v] PEG 2000, 20% [w/v] PEG 400, 0.2 M magnesium chloride, 2.5% 1,4-butanediol. Crystals for the IM structure were obtained in a protein solution consisting of purified enzyme (5.9 mg/mL), 14% [w/v] PEG 2000, 20% [w/v] PEG 400, 0.5 M magnesium chloride, 0.05 M Tris-HCl (pH 8.5), and 3% DMSO. Crystals were grown using the hanging-drop vapor diffusion method with 100 μL of reservoir solution. Diffraction data were collected on beamlines at the Photon Factory, Tsukuba, Japan. All data sets were collected at a wavelength of 1.0000 Å at 95 K. All images were indexed and integrated using the program HKL2000[17], and the data sets were phased with molecular replacement using the program Phaser[18] in the CCP4 program package. The M structure was used as an initial phasing model for the IM structure, whereas the M/NAD/Cys structure was used for the IM/NADP/BU structure. The models were built using the program COOT[19] and refined using Refmac5[20]. The two subunits in the asymmetric unit were refined without crystallographic symmetry restraints. Both the main chain and side chains were clearly identified in the 2Fo–Fc electron density map, and the final difference Fourier maps contained no significant peaks. The programs RAMPAGE[21] and SFCHECK[22] in the CCP4 package were used for stereochemical analyses of all models, for calculation of the RMSD, and for calculation of the average error using the Luzzati plot. The Ramachandran plot analysis of the IM/NADP/BU and IM structures show that 96.5 and 95.6%, respectively, of the residues lie within the most favored region, 3.3 and 4.3%, respectively, of the residues within allowed region and no residues with in disallowed region. Data collection and refinement statistics are presented in Table 2. All figures with illustrated structures were prepared using the program CCP4mg[23] and PDBsum[24]. The coordinates of the IM and IM/NADP/BU structures have been deposited in the PDB under entry numbers 7F4B and 7F4C, respectively.

**Statistics and reproducibility**. The same enzyme was assayed repeatedly more than three times.

**Reporting summary**. Further information on research design is available in the Nature Research Reporting Summary linked to this article.

## Data availability
Data on enzyme activities were deposited on Figshare. All other data are available from the authors upon request.

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

## Acknowledgements

We are grateful to the staff of the Photon Factory for their assistance in data collection, which was approved by the Photon Factory Program Advisory Committee.

## Author contributions

Performed the experiments T.K., E.K., K.W., K.O. and R.K. Designed the experiments and wrote the paper M.G., T.O. and K.Y. All authors read and approved the final manuscript.

## Competing interests

The authors declare no competing interests.
