## [Peer Review File · Communications Biology]

Reviewers' comments:

Reviewer #1 (Remarks to the Author):

Conformational changes in the catalytic region are responsible to heat-induced activation of hyperthermophilic homoserine dehydrogenase

In this manuscript the authors present decent high-resolution crystal structures of apo- and holo-forms of immature homoserine dehydrogenase from hyperthermophilic archaeon *Sulfurisphaera tokodaii* (StHSD). Based on the residue dynamics in the crystal structures, and temperature dependency of enzyme activity, they propose heat treatment to be a method of activation of enzyme activity.

General comments:

Role of temperature in HSD activity has been shown previously in mesophiles, where optimal enzyme activity at 37°C is several times higher than of that at 27°C. So, the increase in enzyme activity as one approaches the optimal growth temperature of the organism is expected. The observed increase in activity with respect to temperature is not substantial enough to claim heat-induced activation is the sole method of "enzyme maturation". Moreover, the protein when purified in mesophiles still shows some basal level activity which presumably could be measured based on the data presented in Fig. 2.

It is unclear as to why the authors believe NADP binding is a feature of immature form of the enzyme. Homoserine dehydrogenase follows an ordered bi-bi kinetic mechanism, in which NADPH binding precedes the binding of L-aspartate-4-semialdehyde at the active site, whereas the product, L-Homoserine, is released first followed by NADP⁺. This suggests that cofactor binding or release are a part of the reaction cycle. It is curious, however, to see that mature and immature forms of the same enzyme prefer NAD and NADP respectively based on the holo-form structures. Because cofactor binding induces conformational changes, and in turn triggers the catalytic activity as evident by varying catalytic site pockets in mesophiles, the difference in the catalytic region conformation that the authors observe between the M and IM form of enzymes could be because of the differences in the nature of ligand (NAD vs NADP).

I feel this study could use more ligand binding studies comparing M and IM forms of enzyme in presence of NAD and NADP, using ITC or differential scanning fluorimetry to determine the cofactor preferences of the enzyme. Also, the role of pH should be probed. As the HSD catalyzed reaction proceeds via dehydrogenation, pH could play a crucial role in the activity. Because the optimal growth temperature and pH for *S. tokodaii* are 80°C and a pH of 2-3, this could be informative.

Specific comments:

Page 5, line 72: Assay data for crude extract activation is not presented. It will be useful to have that as at least supplementary.

Page 6, line 84: It is unclear why eluant should be apo-form. It would be good to have more explanation on this front in methods section and may be some form of absorbance spectra

measurement of protein at 340 nm.

Page 6, lines 86-94: The description that goes with Fig. 2 seems to have some discrepancy. It is hard to see which data denotes "slight activation by additional heating at 343K in the absence of reducing agent". Line 88 says the samples were heat treated without reducing agent before the assay. Lines 90-92 mentions an additional heating step at 343K without reducing agent, but then at 277K there is a heating step in presence of DTT (data for which seems to be missing). This section needs to be rephrased, and Fig. 2 needs to be labelled appropriately, and maybe more data plots need to be included.

Page 7 and 8: It will be useful to have numbers for the conformational changes that are being discussed in Fig 4 and Fig 5. Also, using center of mass of secondary structure features as points to map the movement of the pocket to define open and close form would be helpful. The differences in conformational dynamics in M and IM form of enzymes could also be determined by crystal packing and the surrounding atoms. It is important to rule that out as a cause for conformational change, before assigning these changes to be the reason behind catalytic activity. In addition, temperature dependent (cofactor) ligand binding and activity assays must be studied to corroborate these conclusions. Mutation of the suggested F195, followed by similar cofactor binding and enzyme catalysis assays should be able to throw some light on this.

Page 10, line 152-156: The strength of cofactor binding has not been measured experimentally. It would be pertinent to do that before concluding that the "tight binding of NADP can restrain turnover". The ability to measure HSD catalysis using the assay presented in the paper suggests that the enzyme is binding and unbinding cofactor cyclically as that is a crucial step for hydride transfer between substrate and product, via cofactor.

Table 1: Based on this, it is evident that IM-holo form has some missing residues. It would be interesting to see where they lie and why the resolution for those regions is poor.

Figure 1: Having difference maps for the ligands displayed in the figure could be helpful.

Figure 2: Needs better labeling, also data indicating heating in presence and absence of reducing agent.

Figure 3 and 4: Crystal packing dependent analysis of a similar kind to show these conformational changes are not crystallization artefacts.

Figure 5 and 6: Display distance and movement angle numbers for better understanding.

Reviewer #2 (Remarks to the Author):

Peer review: Conformational changes in the catalytic region are responsible to heat-induced activation of hyperthermophilic homoserine dehydrogenase (manuscript number: COMMSBIO-21-1662)

General comments

The authors determine two crystal structures of the immature *Sulfurisphaera tokodaii* HSD (StHSD), termed IM and IM/NADP/BU, through using a new protein purification method. These two structures were compared to those of the previously measured mature enzymes, designated M and M/NAD/Cys, to reveal the heat-induced activation mechanism. Overall, the current described work is very interesting, and the manuscript is clearly written although there are several grammatical errors. However, I get the impression that the information provided by the authors is limited. A more in-depth characterization and several additional experiments (see specific comments) are required to further improve this manuscript. The author should try to extract as much information as possible based on their available materials and data.

Specific comments for revision

Major comments

1. Page 4, lines 47-48: In previous work, the authors have determined two crystal structures of mature StHSD in apo form, namely PDB 4YDR in the oxidized form and 5AVO in the reduced form. However, in this work, the authors chose the reduced form (5AVO), rather than the oxidized form, as the M structure for extensive structural comparisons with other structures (IM, IM/NADP/BU, and M/NAD/Cys) in the oxidized forms.
2. As we all know, NAD⁺ and Cys are structurally different from NADP⁺ and 1,4-butanediol (BU), respectively. Generally, the ligand-induced conformational changes are tightly associated with the structure of the bound ligand. Therefore, it is entirely possible that the structural comparison between M/NAD/Cys and IM/NADP/BU does not reflect the conformational differences in their pseudo catalytic states. The authors should try to determine the crystal structure of M/NADP/BU or IM/NAD/Cys. If they fails to complete it, the authors should provide a useful discussion or a rational explanation.
3. Page 9, line 141-157. The authors have determined the crystal structure of IM/NADP/BU. However, the authors neither describe in detail the interaction between the enzyme and the bound NADP⁺, nor do they illustrate the binding pattern of NADP⁺ in any figure. They only mentioned that Arg38 shows hydrogen bond interaction with the 2'-phosphate group of NADP⁺ (Page 9, lines 145-147). The description of the NADP⁺ binding mode will facilitate the understanding of the coenzyme preference in StHSD, although it is not the main topic of this study. Therefore, the authors are suggested to complement the description/illustration about NADP⁺ binding in the revised manuscript.
4. The authors have obtained an immature enzyme in apo form via using a Blue Sepharose column. It is very valuable to characterize the general biochemical properties, including kinetic parameters for substrate and coenzymes and effect of pH and temperature on the enzymatic activity (pH and temperature profiles should be illustrated). Of course, their properties should be compared to those

of the mature enzyme to present the heat-induced difference in the biochemical properties.

Unfortunately, the authors only reported the specific activities in the current manuscript.

5. Page 6, line 81: The authors state: "NADP binding is one of the features of immature SthSD".

However, this needs a clear explanation.

Specific comments

1. Table 1: More information should be provided in this Table, such as I/signal, CC1/2, and average B factors for each chain and each ligand.

2. To enhance the reliability of structural data, the electron maps (e.g., SIGMAA-weighted 2Fo-Fc maps) for the ligands of biological interest, such as NADP⁺, should be presented in figures.

3. The authors are suggested to number the secondary structure elements of the structures reported in this work to facilitate the structural descriptions and reader's understanding.

4. Several sentences need to be restructured and rewritten due to the grammatical mistakes or inappropriate expression, such as Page 10, lines 150-152 (It is possible that) and Page 9, lines 144-145 (The hydrogen bond appears to).

5. Page 9, lines 142-143: After visualizing M/NAD/Cys structure by PyMOL, I have noticed that Arg38 can form two hydrogen bonds, rather than just one, with Glu186. Please check it!

6. The universal writing forms for "NAD" and "NADP" are NAD⁺ and NADP⁺, respectively, with plus being superscript, so correct "NAD" to "NAD⁺" and "NADP" to "NADP⁺" through manuscript.

7. Each specific activity reported in this study needs to be expressed as mean \pm standard error, based on at least two independent experiments.

8. Page 6, line 81: Replace "SthSD" with "StHSD".

Response to Comments:

Editorial comments

Please characterize the immature enzyme better by including kinetic parameters for substrate and coenzymes and effect of pH and temperature on the enzymatic activity while comparing it with mature enzyme. Also, kindly do the ligand binding studies as suggested by R#1. Please provide the PDB file for structures.

Answer: Figure 2, Figure 3 and Table 2 showing the pH-dependencies, temperature-dependencies and kinetic parameters, respectively, for the two enzymes were inserted. Since Apparent K_i values of NADP^+ can explain the binding affinity, the values were inserted in Table 2.

Reviewer 1

Role of temperature in HSD activity has been shown previously in mesophiles, where optimal enzyme activity at 37°C is several times higher than of that at 27°C. So, the increase in enzyme activity as one approaches the optimal growth temperature of the organism is expected. The observed increase in activity with respect to temperature is not substantial enough to claim heat-induced activation is the sole method of “enzyme maturation”. Moreover, the protein when purified in mesophiles still shows some basal level activity which presumably could be measured based on the data presented in Fig. 2.

Answer: Since the heat-induced activation was more obvious at higher pHs and temperatures, comparison of pH- (Figure 2) and temperature-dependencies (Figure 3) of the mature and immature enzymes were inserted.

It is unclear as to why the authors believe NADP binding is a feature of immature form of the enzyme. Homoserine dehydrogenase follows an ordered bi-bi kinetic mechanism, in which NADPH binding precedes the binding of L-aspartate-4-semialdehyde at the active site, whereas the product, L-Homoserine, is released first followed by NADP^+ . This suggests that cofactor binding or release are a part of the reaction cycle. It is curious, however, to see that mature and immature forms of the same enzyme prefer NAD and NADP respectively based on the holo-form structures. Because cofactor binding induces conformational changes, and in turn triggers the catalytic activity as evident by varying catalytic site pockets in mesophiles, the difference in the catalytic region conformation that the authors observe between the M and IM form of enzymes could be because of the differences in the nature of ligand (NAD vs NADP). I feel this study could use more ligand binding studies comparing M and IM forms of enzyme in presence of NAD and NADP, using ITC or differential scanning fluorimetry to determine the cofactor preferences of the enzyme.

Answer: Distinct binding affinities of NADP^+ against the immature and mature enzymes were shown by apparent K_i values (Table 2). The much lower K_i value of the immature enzyme suggests that the immature enzyme prefers NADP^+ -binding.

Also, the role of pH should be probed. As the HSD catalyzed reaction proceeds via dehydrogenation, pH could play a crucial role in the activity. Because the optimal growth temperature and pH for *S. tokodaii* are 80°C and a pH of 2-3, this could be informative.

Answer: The temperature (Figure 2) and pH-dependences (Figure 3) were compared between the two forms of the enzyme. As the reviewer suggested, the optimum pH for the growth of *S. tokodaii* is pH 2-3. However it is well known that pH in the cell inside is around neutral pH but not acidic pH. In addition, many NAD(P)⁺-dependent dehydrogenases show generally higher activities at higher pHs in vitro than neutral pH in the cell inside (*in vivo*) because proton (H⁺) is produced accompanying the production of NAD(P)H and the oxidized product.

Specific comments:

Page 5, line 72: Assay data for crude extract activation is not presented. It will be useful to have that as at least supplementary.

Answer: Specific activity of the crude extract was inserted in p. 5 ln. 12-13 .

Page 6, line 84: It is unclear why eluant should be apo-form. It would be good to have more explanation on this front in methods section and may be some form of absorbance spectra measurement of protein at 340 nm.

Answer: Blue Sepharose column binds NAD⁺-binding pocket of the enzyme, and the eluant after several washing of the column must be the apo-form. This explanation was inserted in p. 6 ln. 3-7 of the revised manuscript.

Page 6, lines 86-94: The description that goes with Fig. 2 seems to have some discrepancy. It is hard to see which data denotes “slight activation by additional heating at 343K in the absence of reducing agent”. Line 88 says the samples were heat treated without reducing agent before the assay. Lines 90-92 mentions an additional heating step at 343K without reducing agent, but then at 277K there is a heating step in presence of DTT (data for which seems to be missing). This section needs to be rephrased, and Fig. 2 needs to be labelled appropriately, and maybe more data plots need to be included.

Answer: The results of heat-induced activation was substituted by comparison of temperature and the pH-dependencies and kinetics (Fig. 2 and 3 and Table 2). As the result, all the assay in the manuscript is performed in the absence of reducing agent.

Page 7 and 8: It will be useful to have numbers for the conformational changes that are being discussed in Fig 4 and Fig 5. Also, using center of mass of secondary structure features as points to map the movement of the pocket to define open and close form would be helpful. The differences in conformational dynamics in M and IM form of enzymes could also be determined by crystal packing and the surrounding atoms. It is important to rule that out as a cause for conformational change, before assigning these changes to be the reason behind catalytic activity.

Answer: Explanation of structural differences due to crystal packing was inserted in p.6 ln.17-p.7 ln.2. The numbers of amino residues discussed were highlighted in Figures 4 and 5.

In addition, temperature dependent (cofactor) ligand binding and activity assays must be studied to corroborate these conclusions. Mutation of the suggested F195, followed by similar cofactor binding and enzyme catalysis assays should be able throw some light on this.

Answer: NADP- binding affinity against the immature and mature enzymes was shown by apparent K_i values (Table 2). Temperature-dependent activity was inserted in Figure 2.

Page 10, line 152-156: The strength of cofactor binding has not been measured experimentally. It would be pertinent to do that before concluding that the “tight binding of NADP can restrain turnover”. The ability to measure HSD catalysis using the assay presented in the paper suggests that the enzyme is binding and unbinding cofactor cyclically as that is a crucial step for hydride transfer between substrate and product, via cofactor.

Answer: The strength of NADP-binding was compared and evaluated by the apparent K_i values in Table 2.

Table 1: Based on this, it is evident that IM-holo form has some missing residues. It would be interesting to see where they lie and why the resolution for those regions is poor.

Answer: Location of missing residues was designated with arrows in Figure 1, and discussion about missing residues was inserted in p.8 ln. 2-6 .

Figure 1: Having difference maps for the ligands displayed in the figure could be helpful.

Answer: Difference map was inserted in Figures 1B and 7.

Figure 2: Needs better labeling, also data indicating heating in presence and absence of reducing agent.

Answer: Figure 2 showing heat-induced activation was deleted for clarity.

Figure 3 and 4: Crystal packing dependent analysis of a similar kind to show these conformational changes are not crystallization artefacts.

Answer: Explanation of the packing was inserted in p. 6 ln. 17- p.7 ln. 2.

Figure 5 and 6: Display distance and movement angle numbers for better understanding.

Answer: Distance and movement angle were inserted in the legend of Figure 6.

Reviewer 2

Specific comments for revision

Major comments

1. Page 4, lines 47-48: In previous work, the authors have determined two crystal structures of mature StHSD in apo form, namely PDB 4YDR in the oxidized form and 5AVO in the reduced form. However, in this work, the authors chose the reduced form (5AVO), rather than the oxidized form, as the M structure for extensive structural comparisons with other structures (IM, IM/NADP/BU, and M/NAD/Cys) in the oxidized forms.

Answer: The oxidized (4YDR) form was substituted for the reduced form (5AVO); (p. 4 ln. 7).

2. As we all known, NAD⁺ and Cys are structurally different from NADP⁺ and 1,4-butanediol (BU), respectively. Generally, the ligand-induced conformational changes are tightly associated with the structure of the bound ligand. Therefore, it is entirely possible that the structural comparison between M/NAD/Cys and IM/NADP/BU does not reflect the conformational differences in their pseudo catalytic states. The authors should try to determine the crystal structure of M/NADP/BU or IM/NAD/Cys. If they fails to complete it, the authors should provide a useful discussion or a rational explanation.

Answer: Unfortunately, we failed to determine the structures of M/NADP/BU nor IM/NAD/Cys at this moment. Since both the IM/NADP/BU and M/NAD/Cys share similar structures, the difference of the ligand-binding forms can be small. Discussion of the structures of ligand-binding forms was inserted in p.7 ln. 5-8.

3. Page 9, line 141-157. The authors have determined the crystal structure of IM/NADP/BU. However, the authors neither describe in detail the interaction between the enzyme and the bound NADP⁺, nor do they illustrate the binding pattern of NADP⁺ in any figure. They only mentioned that Arg38 shows hydrogen bond interaction with the 2'-phosphate group of NADP⁺ (Page 9, lines 145-147). The description of the NADP⁺ binding mode will facilitate the understanding of the coenzyme preference in StHSD, although it is not the main topic of this study. Therefore, the authors are suggested to complement the description/illustration about NADP⁺ binding in the revised manuscript.

Answer: Figure 8 was inserted to illustrate the binding, and the sentences in the section "NADP binding" was rephased in p.9 ln. 12-p.10 ln.9.

4. The authors have obtained an immature enzyme in apo form via using a Blue Sepharose column. It is very valuable to characterize the general biochemical properties, including kinetic parameters for substrate and coenzymes and effect of pH and temperature on the enzymatic activity (pH and temperature profiles should be illustrated). Of course, their properties should be compared to those of the mature

enzyme to present the heat-induced difference in the biochemical properties. Unfortunately, the authors only reported the specific activities in the current manuscript.

Answer: The kinetic parameters and dependencies of both pH and temperature were inserted in Table 2 and Figure 2, respectively.

5. Page 6, line 81: The authors state: "NADP binding is one of the features of immature SthSD". However, this needs a clear explanation.

Answer: Experimental results indicating the distinct affinity of NADP⁺ was predicted by the low apparent K_i values which inserted in Table 2.

Specific comments

1. Table 1: More information should be provided in this Table, such as I/sigmaI, CC1/2, and average B factors for each chain and each ligand.

Answer: Additional information was inserted in Table 1.

2. To enhance to the reliability of structural data, the electron maps (e.g., SIGMAA-weighted 2Fo-Fc maps) for the ligands of biological interest, such as NADP+, should be presented in figures.

Answer: The electron map was inserted in Figures 1 and 7.

3. The authors are suggested to number the secondary structure elements of the structures reported in this work to facilitate the structural descriptions and reader's understanding.

Answer: Secondary structure was inserted in Figures 4 and 5.

3. Several sentences need to be restructured and rewritten due to the grammatical mistakes or inappropriate expression, such as Page 10, lines 150-152 (It is possible that) and Page 9, lines 144-145 (The hydrogen bond appears to).

Answer: The sentences were revised in p. 10 ln.7-9 and p. 9 ln. 15-16.

4. Page 9, lines 142-143: After visualizing M/NAD/Cys structure by PyMOL, I have noticed that Arg38 can form two hydrogen bonds, rather than just one, with Glu186. Please check it!

Answer: As reviewer suggested, the second hydrogen bond was found by the program (CCP4) found the hydrogen bond and the bond was shown in Fig. 7.

5. The universal writing forms for "NAD" and "NADP" are NAD+ and NADP+, respectively, with plus being superscript, so correct "NAD" to "NAD+" and "NADP" to "NADP+" through manuscript.

Answer: All the words were corrected.

6. Each specific activity reported in this study needs to be expressed as mean \pm standard error, based on at least two independent experiments.

Answer: Standard error was expressed in the all the specific activities, except for those of the crude extract.

7. Page 6, line 81: Replace “SthSD” with “StHSD”.

Answer: According to the suggestion, it was corrected.

Reviewers' comments:

Reviewer #1 (Remarks to the Author):

Conformational changes in the catalytic region are responsible to heat-induced activation of hyperthermophilic homoserine dehydrogenase

In this manuscript the authors present decent high-resolution crystal structures of apo- and holo-forms of immature homoserine dehydrogenase from hyperthermophilic archaeon *Sulfurisphaera tokodaii* (StHSD). Based on the residue dynamics in the crystal structures, and temperature dependency of enzyme activity, they propose heat treatment to be a method of activation of enzyme activity. In the revised version, the authors report kinetic parameters to show the effect of pH and temperature on enzyme activity and cofactor preference.

Major comments:

1) It is not mentioned in the manuscript at what pH the temperature effects on the specific activity were tested. Nonetheless, the difference in specific activity between mature and immature enzyme is not significant, not at least until the temperature of 353K. More higher temperature readings (>353K) are needed to see if that difference stays and is significant.

2) The difference in specific activity for the same state of the enzyme (either mature or immature) in figure 2 and 3 seems to be substantial. Specific activity of the enzyme in Fig 2 seems to vary between 0-9 U/mg with respect to pH change, while it is varying between 0-110 U/mg with respect to temperature change. In figure 2 it is mentioned that the assay for testing pH variations was performed at 303K, how was this temperature point chosen? Why not test the enzyme at its most efficient (somewhere at 350K) as per figure 3?

3) Assay pHs closer to the isoelectric point of the enzyme are not useful. They can be removed from the figure 2. Comparing the variation in specific activity with respect to pH at two temperatures for example 303K (as shown in figure 2) and at higher temperature (350K) will help in understanding if pH changes contribute to specific activity change in a significant fashion. Similarly, comparing the variation in specific activity with respect to temperature at two pHs, for example pH 8.0 and pH 10.0, will help in understanding if temperature changes contribute to specific activity change in a significant fashion.

4) NADP⁺ apparent affinities have been determined by performing an inhibition assay of sorts to restrain enzyme turnover by forming a NADP⁺-Enzyme complex. This is, perhaps, an indirect measure of cofactor binding, as apparent inhibition constant and ligand dissociation constant are not the same. A better assay that measures direct ligand binding, as suggested earlier will be helpful in supporting the cofactor preference claim. The difference in apparent K_i shown in Table 2, is not substantial to argue that one form prefers NADP⁺ significantly over the other. The difference in the values is within the error or deviation ranges reported.

Minor comments:

1) In figure 2 and 3, it will be useful to see entire curves at these pH or temperature variations

plotted (so that V_{max} is apparent). Because the specific activities at different temperatures or pHs are not being fit to a formula or a curve, the connecting line with valves as symbols could be confusing. A bar plot perhaps is a better form of representing these comparisons if entire curves aren't being included.

2) It might be important to see the difference in specific activity in presence of reducing agent. It would be ideal to perform the same assays reported in Fig 2 and 3 in presence of a reducing agent. Especially because M/NAD/Cys is being compared with IM/NADP/BU in the manuscript. Cysteine is a reducing agent. Studying its effect or a comparable reducing agent's effect on activity in pH or temperature context could provide supporting evidence for the structure-based comparisons in the paper.

3) Use the units for temperature consistently for a same experiment. Either K or C. See lines 88-91 on page 6.

4) Page 8, lines 117-122. Authors mention that a disordered part next to flexible part in catalytic region could be stabilized by heat-treatment. Conventional thermodynamics denotes that heat-treatment destabilizes the disordered or flexible regions in proteins. Perhaps, a better explanation is needed here. Other examples of heat-treatment stabilizing parts in the protein to achieve activation could be helpful for this discussion section.

5) In the absence of an assay measuring direct binding of NADP⁺, the speculation that immature enzyme probably formed a complex with NADP⁺ from E. coli lysate needs to be rephrased. See lines 158-160 on page 10.

6) The methods section for the assay just says 10 mM NADP⁺ was used in the assay buffer. It does not make it clear that NADP⁺ was added to measure the "competitive inhibition by NADP⁺". Please indicate different assay conditions more clearly in the methods.

7) Figure 1. The panel B can be modified to show difference map just for the ligand alone & at a higher magnification than what is shown.

8) pH measurements close to isoelectric point of the enzyme should be removed. It will be useful to note that the difference in activity is not due to loss of enzyme due to pH change.

9) Include a legend for figure 3. Perhaps, also include measurements at a couple of temperature points higher than 353K. Values under 300K could be removed.

10) Please indicate the angle and distance directly on the figure as opposed to just in the legend for figure 6.

11) In figure 7, the difference map for just the ligand & the critical interacting residues could be shown as opposed to entire secondary structure. Also, if figure 8 is not offering new information that is not apparent from figure 7, the authors could consider making a single figure that combines both - display the densities for ligand and interacting residues in the pocket, and display distance of residues from the ligand all in one figure.

Reviewer #2 (Remarks to the Author):

The revised manuscript (COMMSBIO-21-1662A) has addressed most of the concerns that I raised in the first review. However, I still have several concerns regarding the revised manuscript.

Q1: The indispensable methodology descriptions regarding the measurement of kinetic parameters are scarce, which will result in difficulties for other researchers to repeat the observations presented in this manuscript.

Q2 : It is valuable that the authors have measured the kinetic parameters of immature and mature enzymes for coenzyme and substrate. However, these data are only listed in Table 2. I do not see any analysis and discussion about these values in the text. As reported in Table 2, the immature enzyme exhibited a higher affinity for both NAD⁺ and L-Hse compared with the mature enzyme. This is an interesting observation. Could the authors provide structural analyses to support these biochemical data?

Q3. Figure 7. The electron maps for the bound ligands are not clear. Moreover, the type and the contour level of maps should be given in the figure legend.

Q4. The writing forms for "NAD" and "NADP" in the Table 2 should be corrected.

Response to Comments:

Reviewer #1 (Remarks to the Author):

Major comments:

1) It is not mentioned in the manuscript at what pH the temperature effects on the specific activity were tested. Nonetheless, the difference in specific activity between mature and immature enzyme is not significant, not at least until the temperature of 353K. More higher temperature readings (>353K) are needed to see if that difference stays and is significant.

ANSWER: The assay pH (8.0) was inserted in the legend of Figure 3. Specific activities at higher temperatures (353 and 363 K) were inserted in Figure 3.

2) The difference in specific activity for the same state of the enzyme (either mature or immature) in figure 2 and 3 seems to be substantial. Specific activity of the enzyme in Fig 2 seems to vary between 0-9 U/mg with respect to pH change, while it is varying between 0-110 U/mg with respect to temperature change. In figure 2 it is mentioned that the assay for testing pH variations was performed at 303K, how was this temperature point chosen? Why not test the enzyme at its most efficient (somewhere at 350K) as per figure 3?

ANSWER: Since enzyme was partially inactivated after incubation at 353 K for 10 min as shown in Figure S1, assay was performed at temperature as low as possible to prevent inactivation. This explanation was inserted in p.12 ln. 14-18.

3) Assay pHs closer to the isoelectric point of the enzyme are not useful. They can be removed from the figure 2. Comparing the variation in specific activity with respect to pH at two temperatures for example 303K (as shown in figure 2) and at higher temperature (350K) will help in understanding if pH changes contribute to specific activity change in a significant fashion. Similarly, comparing the variation in specific activity with respect to temperature at two pHs, for example pH 8.0 and pH 10.0, will help in understanding if temperature changes contribute to specific activity change in a significant fashion.

ANSWER: Both of the mature and immature enzymes were stable at all the pHs (4.5 - 11) tested for 5 min at 303 K where the assay was performed. This result suggests that the isoelectric point of enzyme has little effect on the activity assayed at 303 K. Since both of the immature and mature enzymes were partially inactivated at 353 K (Fig. S1),

the results at the severe conditions such as at pH 10 or 350 K may be useless due to the instability.

4) NADP⁺ apparent affinities have been determined by performing an inhibition assay of sorts to restrain enzyme turnover by forming a NADP⁺-Enzyme complex. This is, perhaps, an indirect measure of cofactor binding, as apparent inhibition constant and ligand dissociation constant are not the same. A better assay that measures direct ligand binding, as suggested earlier will be helpful in supporting the cofactor preference claim. The difference in apparent K_i shown in Table 2, is not substantial to argue that one form prefers NADP⁺ significantly over the other. The difference in the values is within the error or deviation ranges reported.

ANSWER: We considered carefully the reviewer's comment and thought further discussion based on the structural analysis is difficult since the ligand-binding forms of the immature and mature enzymes share a similar structure. Therefore, we deleted the sentence "These results suggest that the immature enzyme has a higher affinity for NADP⁺ than does the mature enzyme.". Instead, Table S2 was inserted for discussion in p. 10 ln.9-11.

Minor comments:

1) In figure 2 and 3, it will be useful to see entire curves at these pH or temperature variations plotted (so that V_{max} is apparent). Because the specific activities at different temperatures or pHs are not being fit to a formula or a curve, the connecting line with values as symbols could be confusing. A bar plot perhaps is a better form of representing these comparisons if entire curves aren't being included.

ANSWER: Figures 2 and 3 were changed to bar plots.

2) It might be important to see the difference in specific activity in presence of reducing agent. It would be ideal to perform the same assays reported in Fig 2 and 3 in presence of a reducing agent. Especially because M/NAD/Cys is being compared with IM/NADP/BU in the manuscript. Cysteine is a reducing agent. Studying its effect or a comparable reducing agent's effect on activity in pH or temperature context could provide supporting evidence for the structure-based comparisons in the paper.

ANSWER: All the four structures (M, M/NAD/Cys, IM and IM/NADP/BU structures) described in this manuscript are assumed to be oxidized forms based on the distance between two sulfur atoms suggesting formation of disulfide bonds between C303 as described in p.11 ln. 2-9. Cysteine seems to have little effect on the disulfide cleavage, since the M/NAD/Cys structure which was determined by the crystals grown in the presence of 50 mM cysteine is assumed to be oxidized form.

3) Use the units for temperature consistently for a same experiment. Either K or C. See lines 88-91 on page 6.

ANSWER: The temperature was corrected in p. 6 ln. 12.

4) Page 8, lines 117-122. Authors mention that a disordered part next to flexible part in catalytic region could be stabilized by heat-treatment. Conventional thermodynamics denotes that heat-treatment destabilizes the disordered or flexible regions in proteins. Perhaps, a better explanation is needed here. Other examples of heat-treatment stabilizing parts in the protein to achieve activation could be helpful for this discussion section.

ANSWER: Table S1 was inserted to show all disordered residues in the structures and more discussion was inserted in p. 8 ln. 6-11.

5) In the absence of an assay measuring direct binding of NADP⁺, the speculation that immature enzyme probably formed a complex with NADP⁺ from E. coli lysate needs to be rephrased. See lines 158-160 on page 10.

ANSWER: The sentence “These results suggest that the immature enzyme has a higher affinity for NADP⁺ than does the mature enzyme.” was deleted. Instead, Table S2 and discussion in p.10 ln. 9-11 was inserted.

6) The methods section for the assay just says 10 mM NADP⁺ was used in the assay buffer. It does not make it clear that NADP⁺ was added to measure the “competitive inhibition by NADP⁺”. Please indicate different assay conditions more clearly in the methods.

ANSWER: Explanation for the method of inhibition assay was inserted in p. 13 ln. 2-5.

7) Figure 1. The panel B can be modified to show difference map just for the ligand alone & at a higher magnification than what is shown.

ANSWER: Difference map in Figure 1 was modified and the magnification was inserted in the legend.

8) pH measurements close to isoelectric point of the enzyme should be removed. It will be useful to note that the difference in activity is not due to loss of enzyme due to pH change.

ANSWER: The enzymes were stable at all the pHs tested for 5 min at 303 K where the assay was performed. The explanation of pH stability was inserted in p. 12 ln. 14-18.

9) Include a legend for figure 3. Perhaps, also include measurements at a couple of temperature points higher than 353K. Values under 300K could be removed.

ANSWER: Legend and values above 343 K were inserted and those under 300 K were removed in Figure 3

10) Please indicate the angle and distance directly on the figure as opposed to just in the legend for figure 6.

ANSWER: The values are directly inserted in Figure 6.

11) In figure 7, the difference map for just the ligand & the critical interacting residues could be shown as opposed to entire secondary structure. Also, if figure 8 is not offering new information that is not apparent from figure 7, the authors could consider making a single figure that combines both - display the densities for ligand and interacting residues in the pocket, and display distance of residues from the ligand all in one figure.

ANSWER: Figure 7 was improved according to the suggestion.

Reviewer #2 (Remarks to the Author):

The revised manuscript (COMMSBIO-21-1662A) has addressed most of the concerns that I raised in the first review. However, I still have several concerns regarding the revised manuscript.

Q1: The indispensable methodology descriptions regarding the measurement of kinetic parameters are scarce, which will result in difficulties for other researchers to repeat the observations presented in this manuscript.

ANSWER: Explanation of the measurement was inserted in p.12 ln. 12-18 and p.13 ln. 2-5.

Q2: It is valuable that the authors have measured the kinetic parameters of immature and mature enzymes for coenzyme and substrate. However, these data are only listed in Table 2. I do not see any analysis and discussion about these values in the text. As reported in Table 2, the immature enzyme exhibited a higher affinity for both NAD⁺ and L-Hse compared with the mature enzyme. This is an interesting observation. Could the authors provide structural analyses to support these biochemical data?

ANSWER: Since the ligand-binding forms of both immature and mature enzymes share a similar structure, structural difference was discussed based on comparison of their apo-forms in p. 8 ln. 2-5.

Q3. Figure 7. The electron maps for the bound ligands are not clear. Moreover, the type and the contour level of maps should be given in the figure legend.

ANSWER: Figure 7 was corrected for clarity and the contour level was inserted in the legend.

Q4. The writing forms for “NAD” and “NADP” in the Table 2 should be corrected.

ANSWER: Table 2 was corrected.

Reviewers' comments:

Reviewer #1 (Remarks to the Author):

Conformational changes in the catalytic region are responsible to heat-induced activation of hyperthermophilic homoserine dehydrogenase

In this revised version of the manuscript, the authors present high-resolution crystal structures of apo- and holo-forms of immature homoserine dehydrogenase from hyperthermophilic archaeon *Sulfurisphaera tokodaii* (StHSD). Based on the residue dynamics in the crystal structures, and temperature dependency of enzyme activity, they propose heat treatment to be a method of activation of enzyme activity. The authors also report kinetic parameters to show the effect of pH and temperature on enzyme activity and cofactor preference.

Comments:

1. Heat treatment as a method of activation seems unclear based on the assays shown in Figures 2, 3, and S1. The authors claim that at 353K the enzyme was partially inactivated and submit Fig S1 as the evidence. After 353K 10 min treatment, the residual activity was measured at 303K pH 8.0, in the experiment reported in Fig S1. But based on Fig 3 activity at pH 8.0 is poor at 303K as opposed to 353K. So why was the residual activity measured at 303K?

In Fig S1, if untreated enzyme was considered to have activity 1.0, the relative activity of treated enzymes in majority of the cases seem to be between 0.8 and 1.0. Especially, between the pH ranges of 5.0 to 8.0. At basic pH there seems to be a decrease in relative activity of treated enzymes. However, in figure 3, where the enzyme was assayed at pH 8.0 and different temperatures (303K-363K) the activity seems to be increasing with temperature, more so in case of mature enzyme as opposed to immature enzyme. Comparing these two figures, it is unclear how there is increasing activity vs increasing temperature in Fig 3, but not in Fig S1.

Fig 2 shows effect of pH on enzyme activity. Clearly, enzyme shows an increased activity with increasing pH from 7.0 onwards up until 9.0, & then stays relatively same thereafter. The increase in activity is more prominent in mature enzyme. It appears, the enzyme prefers basic pH. Contrasting this, relative activity seems to be lower at basic pH in Fig S1, perhaps because 353K treatment for 10 min destroyed the enzyme. If that is the case, how is heat treatment a mechanism of activation?

Isoelectric point of StHSD enzyme is perhaps about 6.0. All measurements beyond that in Fig 2 shows high activity in mature enzyme vs immature. So, pI might play a role in activity as assayed at 303K, after all, unlike what authors claim in their rebuttal.

The conclusion from the assays seems inconsistent and unclear. Based on the assay results presented, pH seems to play more significant role in enzyme activation as opposed to, or perhaps alongside, the temperature. Based on data presented in Fig S1 and Fig 3, temperature's affect seems to be dependent on pH. Also, Specific activity (U/mg) in Fig 2 and Fig 3 are differing by an order of magnitude or more. For example, compare pH 8.0. If this increase is being attributed to temperature increase (353K in Fig 3 vs 303K in Fig 2) then why aren't all assays measured at higher temperatures. If temperature is increasing activity, why do authors claim partial inactivation when StHSD is

incubated at 353K for 10 min.

2. The values of apparent K_i for NADP+ in Table S2 need more explanation. If the relative activity is same from pH 6.0-8.0, as seen in Fig S1, why is there a huge difference in inhibition of enzyme activity at pH 7.0 vs pH 6.0/8.0. Interestingly, pH 7.0 seems to have far lower K_i than pH 6.0 and pH 8.0, how can this be explained? The pI of the enzyme is about 6.0. At that pH the ligand protein interactions could be compromised, because of enzyme precipitation. This phenomenon seems to be similar for immature and mature enzymes.

3. Based on values reported in Table S2 and Table 2, immature enzyme seems to have more apparent affinity to NADP+/NAD+/L-HSE. There needs to be more discussion on this observation than what is provided. It is unclear how immature enzymes that presumably have more disordered residues than the mature enzymes bind to ligands more tightly. Also, there are no figures showing the kinetics plots corresponding to data presented in Table S2 and 2.

4. On page 4 authors mention that StHSD is shown to be activated via reductive cleavage of the disulfide bond between C303 residues. It is also suggested that all four structures (M, M/NAD/Cys, IM, and IM/NADP/BU) are oxidized forms based on C303 distance. Does that mean that all these structures are of an inactive form StHSD? They also mention that StHSD is shown to be inhibited by cysteine through formation of enzyme-NAD+-cysteine complex with K_i of 11 micromolar. Cysteine perhaps inactivates the enzyme by forming a covalent complex along with cofactor & enzyme. With that as a premise, it will be interesting to see if StHSD gets activated in presence of other reducing agents (due to C303 disulfide break), while is inactivated in presence of Cysteine (which by itself is a reducing agent). This will inform on effect of different kinds of reducing environments on the StHSD activity in addition to pH and temperatures.

Reviewer #2 (Remarks to the Author):

Peer review: COMMSBIO-21-1662B (Conformational changes in the catalytic region are responsible to heat-induced activation of hyperthermophilic homoserine dehydrogenase)

The authors have made corresponding revisions according to the comments raised by me, which addressed the majority of my concerns. There are minor problems as below:

Q1: please correct the writing for coenzyme II as previously described, Page 9, line 148.

Q2: please check Page 10 (lines 157-160 and 167-170). Strange letters!

Reviewer #1 (Remarks to the Author):

1. Heat treatment as a method of activation seems unclear based on the assays shown in Figures 2, 3, and S1. The authors claim that at 353K the enzyme was partially inactivated and submit Fig S1 as the evidence. After 353K 10 min treatment, the residual activity was measured at 303K pH 8.0, in the experiment reported in Fig S1. But based on Fig 3 activity at pH 8.0 is poor at 303K as opposed to 353K. So why was the residual activity measured at 303K?

ANSWER: This manuscript focuses on difference of the activity between the mature and immature enzymes. However, the immature enzyme shows lower stabilities at pH 8.0 and 353 K (Fig. S1) where these enzymes show higher activity. In order to minimize influence of the low stability, activities were assayed at 303 K where both of the enzymes are stable at least for 5 min.

In Fig S1, if untreated enzyme was considered to have activity 1.0, the relative activity of treated enzymes in majority of the cases seem to be between 0.8 and 1.0. Especially, between the pH ranges of 5.0 to 8.0. At basic pH there seems to be a decrease in relative activity of treated enzymes. However, in figure 3, where the enzyme was assayed at pH 8.0 and different temperatures (303K-363K) the activity seems to be increasing with temperature, more so in case of mature enzyme as opposed to immature enzyme. Comparing these two figures, it is unclear how there is increasing activity vs increasing temperature in Fig 3, but not in Fig S1.

ANSWER: Fig. S1 shows residual activities of StHSD after heat-treatment at 353 K for 10 min at various pHs, and the results is distinct from Fig 3 which shows the pH dependencies. This explanation was inserted in p. 6 ln. 86-89.

Fig 2 shows effect of pH on enzyme activity. Clearly, enzyme shows an increased activity with increasing pH from 7.0 onwards up until 9.0, & then stays relatively same thereafter. The increase in activity is more prominent in mature enzyme. It appears, the enzyme prefers basic pH. Contrasting this, relative activity seems to be lower at basic pH in Fig S1, perhaps because 353K treatment for 10 min destroyed the enzyme. If that is the case, how is heat treatment a mechanism of activation?

ANSWER: Most dehydrogenases including StHSD show higher NAD(P)H-forming activities under alkaline conditions where the by-product (proton) is neutralized. Since we monitored the NADH-forming activity of StHSD, the higher activity at alkaline pHs

is adequate. The effect of pH on the activity (Fig. 2) was assayed at 303 K where the enzymes are stable at least 5 min to eliminate the effect of stabilities.

Isoelectric point of StHSD enzyme is perhaps about 6.0. All measurements beyond that in Fig 2 shows high activity in mature enzyme vs immature. So, pI might play a role in activity as assayed at 303K, after all, unlike what authors claim in their rebuttal.

Both of the mature and immature enzymes were stable during the assay at all tested pHs at 303 K in Fig. 2. Thus, Fig. 2 should represent their pH preferences.

The conclusion from the assays seems inconsistent and unclear. Based on the assay results presented, pH seems to play more significant role in enzyme activation as opposed to, or perhaps alongside, the temperature. Based on data presented in Fig S1 and Fig 3, temperature's affect seems to be dependent on pH. Also, Specific activity (U/mg) in Fig 2 and Fig 3 are differing by an order of magnitude or more. For example, compare pH 8.0. If this increase is being attributed to temperature increase (353K in Fig 3 vs 303K in Fig 2) then why aren't all assays measured at higher temperatures. If temperature is increasing activity, why do authors claim partial inactivation when StHSD is incubated at 353K for 10 min.

ANSWER: In order to reveal the difference between the activities of the mature and immature enzymes, the effect of the enzyme stabilities should be minimized. As shown in Fig S1, their stabilities decrease at some pHs at 353 K and their activities cannot be compared properly at high temperature.

2. The values of apparent K_i for NADP⁺ in Table S2 need more explanation. If the relative activity is same from pH 6.0-8.0, as seen in Fig S1, why is there a huge difference in inhibition of enzyme activity at pH 7.0 vs pH 6.0/8.0. Interestingly, pH 7.0 seems to have far lower K_i than pH 6.0 and pH 8.0, how can this be explained? The pI of the enzyme is about 6.0. At that pH the ligand protein interactions could be compromised, because of enzyme precipitation. This phenomenon seems to be similar for immature and mature enzymes.

ANSWER: It is very difficult to discuss pH-dependent K_i values with the obtained structural information, though StHSD may adopted to have low K_i at physiological conditions (pH 7.0). Since both of the immature and mature enzymes are stable at pH 6.0 at 303 K for 5 min as described in the text (p. 13 ln. 207-209), enzyme precipitation cannot occur at the conditions.

3. Based on values reported in Table S2 and Table 2, immature enzyme seems to have more apparent affinity to NADP⁺/NAD⁺/L-HSE. There needs to be more discussion on this observation than what is provided. It is unclear how immature enzymes that presumably have more disordered residues than the mature enzymes bind to ligands more tightly. Also, there are no figures showing the kinetics plots corresponding to data presented in Table S2 and 2.

ANSWER: The more disordered residues in the immature enzyme can induce the enzyme to be relatively more closed-form of the apo-enzyme and decelerate the catalytic turnover. This impediment for the catalytic turnover may decrease K_m values. This explanation was inserted in p. 8 ln. 125-127. Kinetic plots showing data for Table 2 and S2 were inserted in Figure S2 and S3, respectively.

4. On page 4 authors mention that StHSD is shown to be activated via reductive cleavage of the disulfide bond between C303 residues. It is also suggested that all four structures (M, M/NAD/Cys, IM, and IM/NADP/BU) are oxidized forms based on C303 distance. Does that mean that all these structures are of an inactive form StHSD? They also mention that StHSD is shown to be inhibited by cysteine through formation of enzyme-NAD⁺-cysteine complex with K_i of 11 micromolar. Cysteine perhaps inactivates the enzyme by forming a covalent complex along with cofactor & enzyme. With that as a premise, it will be interesting to see if StHSD gets activated in presence of other reducing agents (due to C303 disulfide break), while is inactivated in presence of Cysteine (which by itself is a reducing agent). This will inform on effect of different kinds of reducing environments on the StHSD activity in addition to pH and temperatures.

ANSWER: Since all the obtained crystal structures were the oxidized forms that show relatively lower activity than the reduced forms, the assays were performed in the absence of reducing agents and these enzymes were not treated with reducing agent. Although the mature enzyme was activated by 2 h incubation in 10 mM DTT or 2-mercaptoethanol, the immature enzyme was unstable at the same conditions in the presence of reducing agent. Explanation of the effects of the reducing agents were inserted in p. 11 ln. 175-180.

Reviewer #2 (Remarks to the Author):

The authors have made corresponding revisions according to the comments raised by me, which addressed the majority of my concerns. There are minor problems as below:

Q1: please correct the writing for coenzyme II as previously described, Page 9, line 148.

ANSWER: NADP was corrected to NADP+ in p.10 ln. 151.

Q2: please check Page 10 (lines 157-160 and 167-170). Strange letters!

ANSWER: The strange letters were corrected.

REVIEWERS' COMMENTS:

Reviewer #1 (Remarks to the Author):

The authors have provided explanation to my concerns.